# Optimal diversification strategies in the networks of related products and of related research areas

Aamena Alshamsi[1,2], Flávio L. Pinheiro[1] & Cesar A. Hidalgo [1]

Countries and cities are likely to enter economic activities that are related to those that are already present in them. Yet, while these path dependencies are universally acknowledged, we lack an understanding of the diversification strategies that can optimally balance the development of related and unrelated activities. Here, we develop algorithms to identify the activities that are optimal to target at each time step. We find that the strategies that minimize the total time needed to diversify an economy target highly connected activities during a narrow and specific time window. We compare the strategies suggested by our model with the strategies followed by countries in the diversification of their exports and research activities, finding that countries follow strategies that are close to the ones suggested by the model. These findings add to our understanding of economic diversification and also to our general understanding of diffusion in networks.

[1] Collective Learning Group, The MIT Media Lab, Massachusetts Institute of Technology, Cambridge, MA 02139, USA. [2] Masdar Institute Khalifa University of Science and Technology, Abu Dhabi P.O. Box 54224, UAE. These authors contributed equally: Aamena Alshamsi, Flávio L. Pinheiro. Correspondence and requests for materials should be addressed to C.A.H. (email: hidalgo@mit.edu)

n a world where the probability that a region will enter a new economic activity increases with the presence of related activities, one of the main challenges of regional economic development is to identify the activities that a region should target. Regions may want to focus on activities that leverage local knowledge, but are unlikely to open new development paths. Or they may choose riskier activities, which could be harder to develop but lead to new opportunities. In choosing the latter, there is a question on whether there is a particularly beneficial time to leap into a new territory.

During the last decades a growing literature has shown that the probability that a region will start exporting a new product[1,2], or develop a new industry[3,4], technology[5], or research activity[6,7], increases with the number of related activities present in that location, or with the presence of that activity in neighboring locations[6,8]. These findings, which are evidence of the social, technological, geographic, and economic constraints to knowledge diffusion, have been modeled using network methods[9] that allow scholars to connect related activities and by new datasets that summarize patterns of co-location[1] and co-production[3,5,7], as a way to measure the relatedness of activities. This literature has given rise to a nuanced understanding of the empirical path dependencies that shape economic development, but has left unanswered questions about the optimal strategies needed to traverse these development landscapes.

The literature on network diffusion, on the other hand, has explored how the structure of networks, and the seeding of epidemics or information, affects diffusion[10–13]. Hence, it presents a good point to start thinking about strategies for cases when knowledge diffusion is constrained by the relatedness of activities. An important distinction within this networks' literature is the difference between simple and complex contagion[14–16]. Simple contagion involves situations where transmission requires contact with a single individual[17]. Complex contagion involves reinforcement by multiple sources[14–16], and hence, is a more accurate representation of development processes involving knowledge diffusion, where the probability of success is known to increase with the presence of related activities[1–8]. Therefore, we can create models that optimize knowledge diffusion by building on the idea of complex contagion[14–16].

Here, we formalize the problem of identifying optimal economic diversification strategies as a problem of strategic diffusion in the presence of complex contagion. We explore this problem for simple and complex network topologies, finding that strategies that minimize the total time needed to diversify an economy target highly connected nodes at an intermediate step of the process. We compare the strategies with the empirical behavior observed for countries in the diversification of their exports and research publications, finding that this empirical behavior is similar to the strategies suggested by our theory.

## Results

**Path dependencies in networks of related activities.** To identify strategies that optimize knowledge diffusion we first empirically characterize path dependencies, and then use this characterization to simulate development through various strategies. Fig. 1a, b show the empirically determined networks of related products (a) and related research areas (b). Products are connected if they are likely to be co-exported[1] and research areas are connected if the same scholars are likely to have published in both of them[7] (See Supplementary Notes 5 and 6 and original papers for data and methods). These networks define path dependencies for the development of new exports and research activities. Figure 1c, d shows, respectively, the path dependencies for the network of

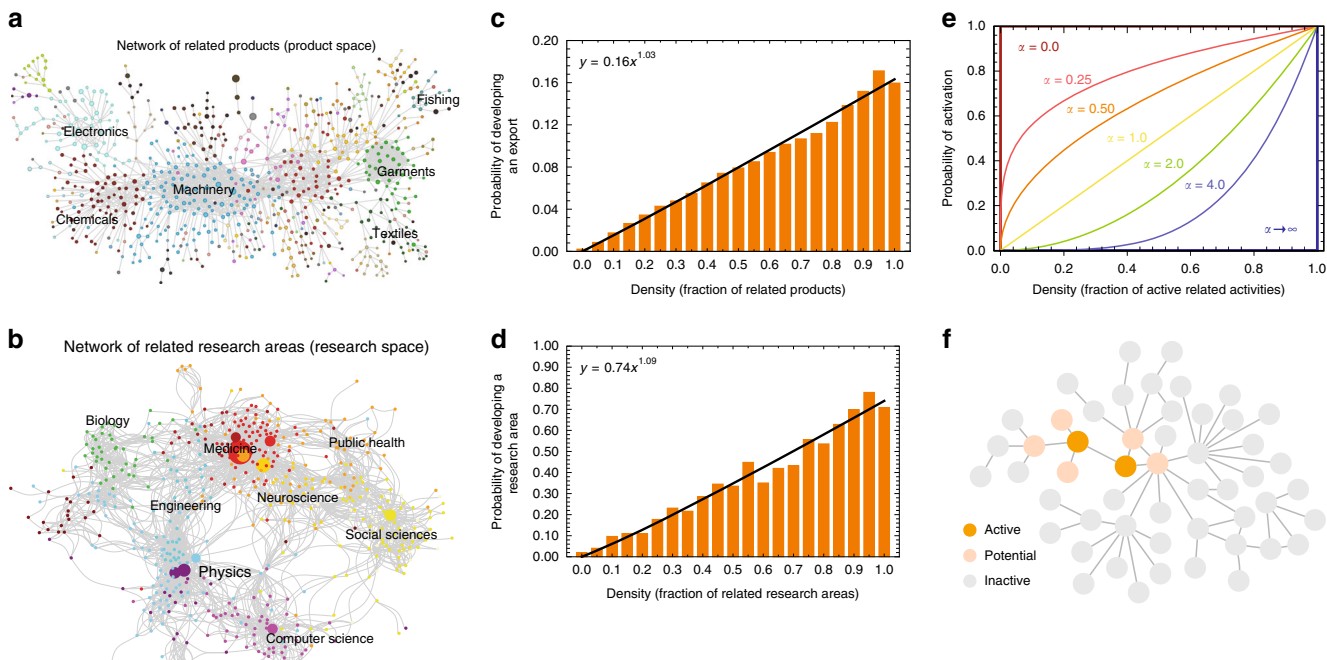

**Fig. 1** Modeling the empirical development of exports and research areas. **a** Network of related products or product space (from ref. [1]). **b** Network of related research areas or research space (from ref. [7]). **c** Probability that a country's export per capita in a product surpasses 25% of the world's average as a function of density: the fraction of related products already exported by that country. **d** probability that the number of per capita publications of a country in a research area becomes larger than the world's average as a function of density: the fraction of related areas where that country already participates in **e**. We model the behavior observed in **c** and **e** by using a power-function for the probability that an activity will be developed by a location (Eq. 1). Here we show the behavior of this power function for different values of $\alpha$. **f** We model diffusion by assuming nodes in a network can be in three possible states: inactive (gray), potentially active (peach), and active (orange). Potentially active nodes are connected to active nodes but are not yet active

related products and the network of related research areas. Here we can see that the probability that a country will become a significant exporter of a product in a 4 year period increases with the fraction of related products already exported by that country. By the same token, the probability that a country will enter a new research area in a 3 year period increases with the fraction of related research areas where that country is actively publishing in (Fig. 1d).

We can model this behavior by noticing that in both cases the probability that a country will start exporting a new export, or become active in a research area, increases as a power of the fraction of related activities present in that location ($y = ax^b$). This empirical law means that the cost of developing a related activity is much lower than the cost of developing an unrelated activity, since the probability of success is much higher for related activities. For the product space and the research space this power function has a slightly super-linear form, which is respectively: $y \approx 0.16x^{1.03}$ for the product space and $y \approx 0.74x^{1.09}$ for the research space (See SI for more information).

**Modeling strategic diversification**. We can bring these empirical findings into a theoretical model by modeling the probability that a location will develop a new activity as (Fig. 1c, d):

$$p_i = B \left( \frac{\sum_{j=1}^{k} a_{ij} M_j}{k_i} \right)^{\alpha} \qquad (1)$$

where $a_{ij}$ is the adjacency matrix connecting related activities $i$ and $j$ ($a_{ij} = 1$ if there is a link and 0 otherwise), $M_j$ is a memory vector indicating if activity $j$ is active ($M_j = 1$) or inactive ($M_j = 0$) (the country is an exporter of a product, or participates in a research area), $k_i$ is the total number of activities related to activity $i$ (whether they are active or not) ($k_i = \sum_j a_{ij}$), $B$ is the probability of activation when all related activities are present, and $\alpha$ is a parameter that helps us adjust the strength of path dependencies. For intuition, $\alpha = 0$ means that the probability of activating a node is the same for nodes with many or few related activities (no complex contagion or no path dependence), $\alpha = 1$ means that the probability of activating a node is linearly proportional to the number of related activities present in a location, and $\alpha > 1$ means that the probability of activating a node increases concavely with the number of related activities present in a location. Both the product space, and the research space, exhibit a behavior that is slightly super linear ($\alpha > 1$ as shown in Fig. 1c, d).

We then use this model to identify the optimal development in a network where activities (products or research areas) can be in one of three states: active (A), potentially active, (P) or inactive (I) (Fig. 1e). Potentially active nodes—nodes with non-zero activation probability—can be activated with a probability following our empirically informed model (Eq. 1).

Since developing activities is costly, we model diffusion sequentially, by choosing one potentially active node as an activation target at each time step (the target). A strategy, therefore, can be described as an ordered sequence of activation targets. Hence, the strategic problem that we need to solve is to identify a sequence of targets that minimizes the total time needed to activate all nodes in a network. Recently, increasing economic diversity has become an explicit economic development goal for multiple countries. Saudi Arabia[18], Peru[19], Chile[20], and Indonesia[21], are just a few examples that have made increasing the diversity of their economies an explicit development goal. Certainly, one could focus on other objective functions (such as maximizing the contribution of exports to GDP, minimizing

inequality[22], or the carbon footprint[23] of a country's export structure). For now, we leave the extension to other objective functions as a future exercise and focus on minimizing the total time needed to activate all of the nodes in the network.

Also, we do not consider strategic interactions (situations in which multiple diffusion processes are competing for the same nodes in the network). While we acknowledge that the problem of strategic interactions is an interesting one, we use instead what is a standard assumption in economic development. This is that world markets are relatively large compared to domestic markets. This assumption is validated by the data, since—on average—new entrants represent less than 1 percent of the global market of a product. So the real competition that new entrants face is not with each other, but with large incumbents. Hence, we focus on exploring solutions for the problem of strategic diffusion in the case of a single objective function (diversification) and in the absence of strategic interactions. Yet, even in this simple case, the problem of finding optimal sequences for complex contagion is computationally expensive, so we make simplifications that make the problem tractable. Going forward, we call the sequences that minimize the total time needed to activate the network an optimal diversification strategy.

We begin by solving the model for three topologies: a wheel network (a network with a central hub surrounded by a ring of nodes) (Fig. 2a), a generalized wheel network (a network with multiple hubs connected to a large number of nodes in a random network) (Fig. 2b), and a scale-free network constructed using the configuration model[24] (Fig. 2(see Fig. 1c)). We use the wheel network to develop our basic intuition, and then, use the more complex network topologies to show that the intuition developed in the wheel network generalizes to more complex cases. These three networks have heterogeneous degree distributions where few nodes have high degrees while many nodes have low degrees. Their structures are similar to the network structures of research space and product space so we can readily assume that the most effective strategies in these three networks are the most effective ones in the two spaces. Then, we bring the theory to the data by returning to the network of related products and the network of related research activities.

We benchmark our strategies using five simple diffusion strategies: random strategy: where we target potentially active nodes at random; high-degree strategy: where we target the potentially active node with the highest degree; low-degree strategy: where we target the potentially active node with the lowest degree; greedy strategy: where we target the node with the highest probability of activation; and majority strategy: where we target the potentially active node with the highest number of active neighbors (which can still have a small probability of activation when that node has many neighbors). The motivation for each of the five strategies is as follows. On the one hand, we have low-hanging fruit strategies, such as greedy and low degree. These are strategies that target easy to activate nodes, but fail to consider the future strategic value of these nodes (how many nodes they will help activate in the future). On the other hand, we have more ambitious strategies, such as majority and high degree, which target nodes that could be difficult to activate early on, but that can help activate other nodes later in the process. The high-degree strategy is particularly interesting as a benchmark because it is a common heuristic in problems of simple contagion[11,25]. Finally, the random strategy allows us to describe what to expect in the absence of strategic choices, and hence, is an important null benchmark that any sensible strategy should beat.

Consider a wheel network populated by $Z$ nodes (Fig. 2a). A wheel network has a central hub, which is connected to all nodes, and a ring of $Z-1$ nodes that are connected to two neighbors and the hub. The wheel network is particularly instructive because the

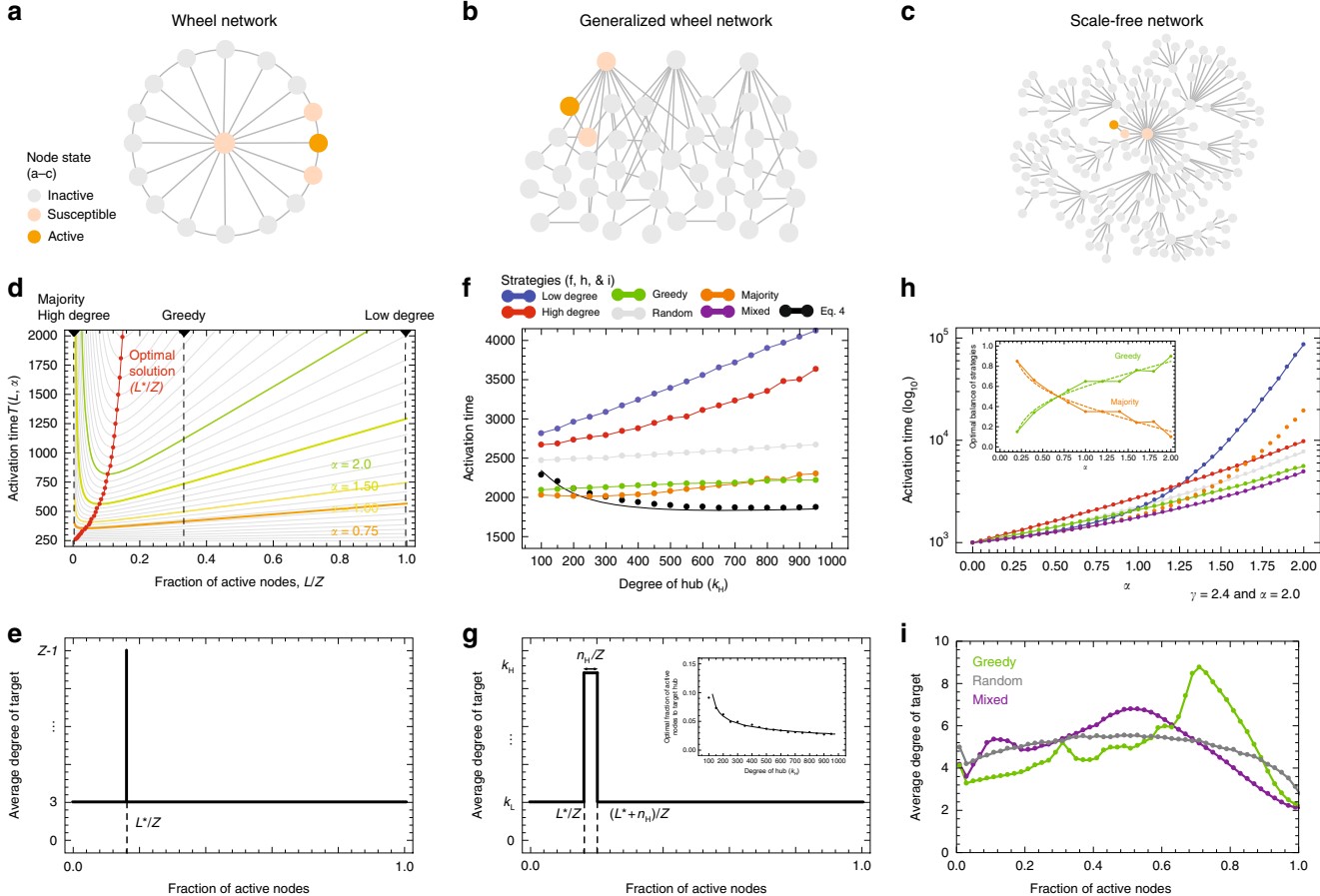

**Fig. 2** Strategic diffusion in networks. Network models. **a** Wheel network, **b** generalized wheel network, and **c** scale-free network. **d** Time needed to activate all nodes in the wheel network as a function of the time in which the hub is targeted, with time expressed as the fraction of active nodes (Eq. 2). Optimal time is indicated with the red line. **e** optimal strategy for the wheel network described as the degree of the nodes that is optimal to target for each fraction of the total nodes that have been activated. **f** Total time needed to activate a generalized wheel network as a function of the degree of hubs for the five simple strategies and for the solution of Eq. 4 (in black line, black dots represent a simulation). **g** optimal strategy for the generalized wheel network described as the degree of the nodes that is optimal to target for each fraction of the total nodes that have been activated. Inset shows optimal targeting time as a function of the degree of the hubs. **h** Total activation time to activate a scale-free network for the five pure strategies and the optimal mixed strategy as a function of the parameter modeling the importance of relatedness $\alpha$. Inset shows optimal mix of greedy and majority strategies as a function of $\alpha$. **i** Dynamic strategies for the scale-free networks obtained via numerical simulations for the greedy, random, and the optimal mixed strategy. The mixed strategy targets hubs at an intermediate level

problem of strategic diffusion reduces simply to that of choosing when to target the hub. In the wheel network, the probability of activating peripheral nodes ($p_i = (1/3)^\alpha$) does not change unless the hub is active. After the hub is active, the probability of activating peripheral nodes grows to $(2/3)^\alpha$.

We start with a single active peripheral node. The greedy strategy in this case is to always develop the activity with the highest probability of success, and hence, to activate 1/3 of the peripheral nodes before targeting the hub. The majority strategy would target the hub after one peripheral node is infected, and would be almost identical to the high-degree strategy. The low-degree strategy would target the hub last.

We can obtain an optimal strategy by leveraging the symmetry of the wheel network to write an equation for the average total time needed to activate the full network.

Let $L$ be the time when we target the hub, measured as the number of peripheral nodes that have been activated. Also, note that the expected time $t$ required to activate a node with activation probability $p$ is $t = 1/p$. Then, the total time required to

activate the entire network will be equal to:

$$T(L, \alpha) = 3^\alpha(L-1) + \left(\frac{Z-1}{L}\right)^\alpha + \left(\frac{3}{2}\right)^\alpha(Z-2-L) + 1 \quad (2)$$

where $3^\alpha$ is the time required to activate each of the first $L-1$ nodes, $((Z-1)/L)^\alpha$ is the time required to activate the hub, and $(3/2)^\alpha$ is the time required to activate all remaining peripheral nodes after activating the hub, except for the last one which takes one unit of time.

Figure 2d shows the total time needed to activate the entire network as a function of $L$ and $\alpha$ ($T(L,\alpha)$). When $\alpha$ approaches zero, relatedness does not matter and therefore, activation times do not depend on when you target the hub. As $\alpha$ increases, and we dial in the importance of relatedness, an optimal solution emerges. For $\alpha = 1$ the time is minimized when about 7% of all nodes are active. For $\alpha = 2$, the optimal time to target hubs is when 10–15% of the peripheral nodes are active. This optimal window is sooner than what we would obtain from the greedy

strategy. All other strategies (majority, low degree, high degree, and random) perform poorly. Also, we note that the optimal window is asymmetric. Since the functional form of Eq. 2 (Fig. 2d) is such that it decays fast near 0, and rises slowly after the optimal. This tells us that in an uncertain situation, it is less risky to be one step late than one step too early.

Figure 2e shows a representation of the optimal strategy for the wheel network. Here, the x-axis shows the fraction of the network that has been activated, and the y-axis shows the degree of the nodes that is optimal to target at that time. The optimal value ($L^*$) can be obtained by setting $dT/dL = 0$:

$$L^*(\alpha) = (Z-1)\left(\frac{((3/2)^\alpha + 3^\alpha)(Z-1)}{\alpha}\right)^{-\frac{1}{\alpha+1}} \quad (3)$$

The wheel network model has four important implications. First, it tells us that the optimal window of time to target the hub activity is lower than we would expect from the greedy strategy that focuses only on the most related activity $L^*_{\text{optimal}}/Z < L^*_{\text{greedy}}/Z = 1/3$. Second, it tells us that targeting hubs too early is a strategy that performs poorly. Third, it shows us that the difference between the optimal solution and the sub-optimal solutions increases with $\alpha$, meaning that the value of using the right strategy is small when relatedness is not important ($\alpha \ll 1$), but large when relatedness is substantial ($\alpha \geq 1$). Finally, we find the optimal target values increase with $\alpha$, meaning that we need to target hubs later in the process when the importance of relatedness is stronger.

Next, we extend these results to a generalized wheel network and a scale free-network (See SI Section 2.1 for more details). We call a generalized wheel network a network with $m$ hubs connected to $k_H$ low degree nodes that form themselves a random network with average degree $k_L \ll k_H$. The total number of low degree nodes is $n$. (Fig. 2b). This allows us to generalize the idea of the wheel network and write an equation for the total activation time as:

$$T(L,\alpha) = \sum_{j=2}^{L}\sum_{i=0}^{k_L-1}\binom{n-2}{k_L-1}^{-1}\binom{j-1}{i}\binom{n-j-1}{k_L-1-i}\left(\frac{mk_H+k_L}{n(1+i)}\right)^\alpha$$
$$+ m\sum_{i=0}^{L-1}\binom{L}{i}\binom{n-L-1}{k_H-i}\binom{n}{k_H}\left(\frac{k_H}{i+1}\right)^\alpha$$
$$+ \sum_{j=L+1}^{n}\sum_{i=0}^{k_L-1}\binom{n-1}{k_L}^{-1}\binom{j}{i}\binom{n-j-1}{k_L-i}\left(\frac{mk_H+nk_L}{mk_H+in}\right)^\alpha \quad (4)$$

In Eq. 4, the first sum accounts for the activation time of the initial $L$ low degree nodes, the second term for the activation of the $m$ hubs, and the third term accounts for the activation time of the remaining $n-L$ low degree nodes. Since this expression does not have a closed form solution, we explore it numerically. For more information about the equation, see SI Section 3.2.

Figure 2f compares the predictions of Eq. 4 with the results obtained for numerical simulations of the five strategies described above: random, high degree, low degree, greedy, and majority. We vary the heterogeneity of these networks by increasing the connectivity of hubs (increasing $k_H$). For illustration purposes, we consider generalized wheel networks with 1000 nodes (10 hubs and 990 low degree nodes) although the results are not too sensitive to the number of hubs.

Figure 2f shows that the difference between solutions is larger for more heterogeneous networks, meaning that choosing the right dynamic strategy is more important when networks are

heterogeneous. In fact, Eq. 4 provides a strategy that activates the whole network in 80 percent of the time of the greedy and majority strategies, and in 70 percent of the time required using the random strategy, for the most heterogeneous networks. Figure 2g shows the optimal strategy by plotting the connectivity of nodes that is optimal to target at each step. The optimal time to target hubs in a generalized wheel network as a function of their degree is shown in the inset.

Finally, we explore diffusion in scale free networks[26] (Fig. 2c). Scale-free networks are heterogeneous networks characterized by a power-law degree distribution, meaning that the probability that a node will be connected to $k$ other nodes follows a distribution of the form $P(k) \approx k^{-\gamma}$ (See SI Section 2.2 for more details). Since we do not have an analytical solution for the case of scale-free networks we explore the problem numerically. Figure 2h compares the performance of the five aforementioned strategies (random, low degree, high degree, greedy, and majority) with the performance of a strategy that mixes the greedy strategy (with probability $p$) and the majority strategy (with probability $1-p$) to minimize diffusion time. This mix of strategies beats all benchmarks, and performs relatively better when relatedness is more important (higher $\alpha$). The inset of Fig. 2h shows the optimal mix of greedy and majority needed to minimize the total diffusion time for each $\alpha$. Figure 2h characterizes the strategy identified by this optimal mix by showing the average degree of the nodes targeted at each diffusion step. Once again, we observe that the optimal strategy targets high degree nodes earlier than a strategy focused on the nodes with the highest probability of activation (greedy).

**Empirical validation.** We now can bring the theory to the data by looking at the average strategies followed by countries in the development of new exports and research areas. Like we did for our models, in Fig. 2e, g, i, we plot for both of these networks the average connectivity of the nodes developed by a country as a function of the number of active nodes (the level of diversification of exports or diversification in research activities). We then compare the empirically observed behavior with an optimal strategy obtained by using a combination of greedy and majority strategies. In both cases, we follow the empirically observed activation probabilities shown in Fig. 1.

Figure 3a shows the average connectivity in the product space of a country's new exports as a function of its level of diversification—fraction of all exported products—when the country started exporting each product with comparative advantage. The empirical behavior matches the predictions of the model qualitatively, in that both the empirical curve and the model exhibit an inverse-U relationship. That is, countries tend to jump to more connected products at an early but intermediate level of diversification, as it would be suggested by our theory of strategic diffusion. Yet, compared to the model, countries tend to overshoot for hubs early in the process, and also, slightly undershoot at intermediate steps. A similar behavior is observed for countries developing new research areas in the research space. Here, the optimal of the simulation is close to what we expect from the theory.

These results show that the process of diversification of countries in the product space and research space is—on average—close to what we should expect from a model of strategic network diffusion focused on minimizing the diversification time.

## Discussion

Scholars have long observed that similar productive activities agglomerate[27]. While many mechanisms have been proposed in the past to explain such agglomerations, knowledge diffusion has

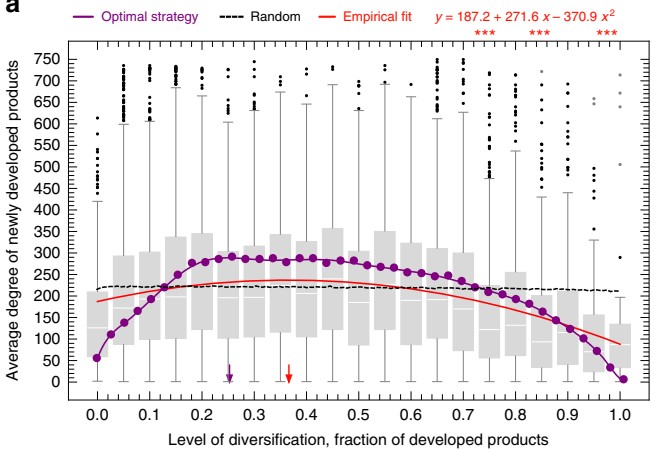

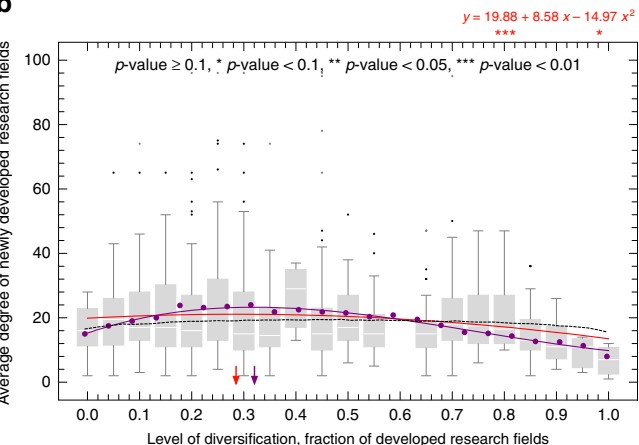

**Fig. 3** The dynamics of economic and research diversification. **a** Boxplot showing the average connectivity of newly developed products in the product space as a function of the diversification level of the countries entering that product. The red line shows the fit for the empirical values (the results of the statistical test for the significance is shown at the top) and the purple line shows the optimal mix of greedy and majority strategies obtained via numerical simulations. The black line shows the null model baseline which uses the random strategy. **b** Boxplot showing the average connectivity of newly developed research areas in the research space as a function of the diversification level of the countries entering that area. The red line shows the fit for the empirical values (the results of the statistical test for the significance is shown at the top), and the purple line shows the optimal mix of greedy and majority strategies obtained via numerical simulations. The black line shows the null model baseline which uses the random strategy. There is a bin missing in **b** (at diversification = 0.6) because there were no events with that value on the observation period. In both **a** and **b** boxplots show the interquartile range inside the box, and have whiskers that extend to the top 95% and bottom 5% of the distribution

emerged as one of the main drivers of related activities at multiple scales, from cities, to regions[3], to countries[1]. The main finding supporting this idea is the fact that the probability of entering an activity increases with the number of related activities present in a location, or in nearby locations, probably because similar activities require similar inputs (which are most likely, knowledge based)[1–8,28–31]. Yet, despite the prevalence of this relationship, little is known about the strategies that are optimal to maximize diversification when knowledge diffusion is constrained by a network of related activities.

Here, we explore strategies to maximize knowledge diffusion in situations where the probability that a region will enter an activity increases in the presence of related activities. Of course, this is not a simple problem, so we present a stylized version of it focused on maximizing diversification for a few theoretical networks and two previously published networks of relatedness: the product space[1] and the research space[7]. Yet, one could extend these ideas to other objective functions, such as total contribution of exports to GDP, economic complexity[32], or the carbon intensity of products[23]. To do this, one would assign each node to a value (e.g., its potential contribution to GDP), and would search for a strategy that finds the sequence that maximizes an aggregation of these values. In our case, we focus on diversification as an objective function because it represents a convenient baseline, one where all nodes have the same value. Yet, other functions could be explained as well.

Here, we explore this problem analytically for simple network structures, and numerically for complex networks, to show that the strategies needed to maximize diversification in the presence of related activities are dynamic, and require focusing not only on which activities to target, but on when to target highly connected activities. Our findings suggest that efforts to target highly connected activities and research areas should be optimal at an intermediate, and relatively low, level of diversification. These findings improve our understanding of strategies that can help maximize knowledge diffusion, and also, add to our general understanding of diffusion in networks.

**Data availability**. The details of all data and methods used are given in Supplementary Notes 5 and 6 and original papers[1,7].

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

## Acknowledgements

This work was funded by the Cooperative Agreement between the Masdar Institute of Science and Technology (Masdar Institute), Abu Dhabi, UAE and the Massachusetts Institute of Technology (MIT), Cambridge, MA, USA—Reference 02/MI/MIT/CP/11/07633/GEN/G/00. Also, this work was supported by the Center for Complex Engineering Systems (CCES) at King Abdulaziz City for Science and Technology (KACST) and the Massachusetts Institute of Technology (MIT) and the MIT Media Lab consortia. We thank Miguel R.Guevara for help with the data. We also thank Tarik Roukny for the useful discussions and suggestions.

## Author contributions

Conceived and designed the experiments: A.A., F.P., and C.H. Performed the experiments: A.A., F.P. Analyzed the results: A.A., F.P., and C.H. Wrote the paper: A.A., F.P., and C.H.

## Additional information

**Competing interests:** The authors declare no competing interests.

