## [Peer Review File(PDF 308 kb) · Nature Communications]

Reviewers' comments:

Reviewer #2 (Remarks to the Author):

The paper aims at going in depth in path dependent processes of knowledge diffusion by finding optimal strategies on new activities introduction, when activities are considered being related each other. Two key points are developed. First, the authors stress on the role of various structures of technological and scientific relatedness on the diffusion patterns, from wheel to scale-free networks. Second, they confront analytical and numerical results to real large dataset on knowledge and product spaces. Undoubtedly, the paper constitutes an original research that offers something new that was clearly missing in the growing research field of relatedness, diversification and economic growth, i.e. some theoretical and analytical foundations, as well as some strategic considerations on how and when to diversify at the country level.

The methodology does not suffer from criticisms and the balance between analytical and numerical simulations is perfectly justified. The data used for confronting results to real data on product and research spaces are well-known and constitute a reference in the community.

Then I consider the paper meets the quality standards of the journal. Only two remarks can be done, and could be taken into consideration for the final publication. First, the authors can suggest intuitions or announce further developments (and not demonstration) related to other network structures. The three structures considered in the paper are in a sense very limited. For instance, the structure of relatedness in science as well as in industry can present very different level of assortativity according to the different cultures in different countries. Considering assortative and non-assortative structures can help better understanding knowledge diffusion and different patterns of diversification. Second, I think that the conclusion presents some confusing arguments. In particular, the reference to Marshall and agglomeration (and then, implicitly to Krugman on increasing returns and economic geography) seems to be unfitting with the topic under study. The product and knowledge spaces on which the authors work are completely disconnected from the spatial structure of countries. Exports and publications are simply considered as national aggregates, without any considerations on particular spatial structures such as cities or clusters. Then, the conclusion could be inappropriate for geographers. I would suggest making some changes in the conclusion, by referring more on researches related to economic policy in industry and science (what kind of public incentives to better target diversification strategy?)

Pr. Jerome Vicente, University of Toulouse - France

Reviewer #3 (Remarks to the Author):

The authors address the interesting problem of knowledge diffusion as a complex contagion, and examine the effects of interdependency on optimal diffusion time. The overall idea is well-grounded and offers an important direction for research on productivity and knowledge networks.

The authors could help to improve the clarity and reach of their project by making some changes in the exposition:

- 1) They would do well to foreground the important point that for knowledge diffusion, as for all complex contagions, the role of hubs is determined by interdependencies with the peripheral nodes. This study demonstrates how these interdependencies shape the strategic role of hubs in accelerating knowledge diffusion (aka, “activation”).

- 2) When α is first introduced, it would be useful if the authors provide empirical intuitions for situations where α is low, medium and high. This will add clarity to the theoretical and numerical analyses, which depend on this parameter.

- 3) There is an abrupt transition in the exposition of the manuscript on the top of page 3, where the logic of the argument transitions from describing the interdependences in activation dynamics in the empirical production and publication data, to framing the problem of when to activate a central node. There is a bit of a skip here –it does not follow naturally for the reader from the first two pages that “the strategic problem that we need to solve is to identify a sequence of targets that minimizes the total time needed to activate all nodes in a network.” Subsequently, it is clear that that the authors’ goal is to know when to activate areas of production in order to further increase (i.e., spread) production. However, the presentation here would benefit from a brief explanation of who faces this strategic problem, and what kinds of decisions actors are expected to make (e.g., activating hubs in the production network). This motivation would be helpful for connecting the theoretical and numerical analyses with the concluding data on empirical targeting decisions, and the lessons that these results have for actors who make these decisions. As the authors struggle a bit to contend with this point in the final paragraph, it would be useful to anticipate this issue a bit earlier in the paper.

4) It would be useful if the authors would provide some motivation for the five simple strategies they test. It is clear why they choose them: they use several familiar targeting strategies that are more or less representative of known techniques; however, it would be helpful if the authors could give a sense of the larger space that these targeting strategies explore. This would circumscribe the limits of their numerical explorations vis-à-vis their theoretical findings.

Additional minor points:

i) The text has numerous grammatical and syntactic errors, which can be easily fixed. The intelligibility of the text would be greatly improved by editorial treatment.

ii) This is not necessary, but the authors might consider using 'activity diffusion' instead of 'knowledge diffusion' since, as they discuss, there are many reinforcing factors (empirical path dependencies) that create complexity in the activation of nodes in a productivity network, and activation is not necessarily due to the spread of any particular kind of knowledge.

Overall, the authors make a useful connection between several streams of research on complex contagions, knowledge diffusion, network structure, and productivity.

Reviewer #4 (Remarks to the Author):

The paper analyses diffusion strategies in networks of related activities. A complex contagion model is developed to identify which activity should be targeted at each step. In the model, the probability that an agent will develop a new activity increases as a power function of active related activity with exponent α , as described in Equation 1. The optimization problem is to minimize the total time needed to activate all nodes in the network (i.e. to develop all activities). An optimal solution to the problem is obtained for a wheel network (Eq. 3) whereas for other network topologies (generalized wheel and scale-free networks) numerical results are provided. The optimal strategy mixes the greedy and the majority strategies. The optimal mix favors majority for low values of α and the greedy strategy when α is high enough. Results are validated by looking at the strategies followed by countries in the development of new exports and research areas.

I have a few major concerns about the paper and there are some other minor issues that should be addressed by the authors, especially in the empirical part of the manuscript. My comments are as follows:

1) Strategic interaction is not considered. Agents develop a new activity if they outperform rivals ($RCA \text{ per capita} > 1/4$ and $RCA \text{ per capita} > 1$ for research and exports, respectively). RCA depends on the strategies of other agents. As it is evident in Eq. 14 in the appendix, RCA of country c is a function of exports of other countries c' . As described in Fig. 1(f) a potentially active product is connected to active nodes in the product space. By construction, connections are present when a sufficiently large number of countries jointly export active and potentially active products, with $RCA > 1$. This implies that the optimal strategy of country c depends on the strategy of other agents who are likely to develop the same product. For instance, if all of them target the same potentially active node by increasing per capita exports to the same extent, their efforts are likely to cancel out. Neglecting to consider strategic interaction in this setting is a major limitation of the paper. More in general, strategic interaction is core in game-theoretic competition models widely applied to international studies. Also, the way in which the RCA thresholds are fixed is totally arbitrary. A justification is needed of why we should consider as a relevant transition when RCA goes from lower than 0.25 to an RCA greater than 0.25 in the product space.

2) Diversification is not the objective. In the paper, the authors implicitly assume that all activities (products, publications) are of the same value. Evidently, this is not the case. Thus, the objective function is questionable. It is much more sensible to assume that countries' objective in trade is to maximize the contribution to GDP (or the trade balance) per capita (total exports per capita – total imports per capita). Total exports is the sum across all exported products (extensive margin) of the dollar value of exported goods (intensive margin). Thus, countries can develop strategies to export less goods but at a higher value. Indeed, it is well known that most developed countries export less products than mid-income countries, and no country exports all goods. The same trade-off applies to scientific production (less papers of higher impact). In sum, we cannot take for granted that maximal diversification is good. Indeed, policies in Europe and elsewhere are meant to pursue smart specialization, not necessarily as a way to minimize the time to maximal diversification.

3) Empirical results are weak. The authors claim that countries follow strategies that are slightly sub-optimal. Having said that we cannot discard the possibility that countries are just more sophisticated and they follow an optimal strategy which considers (1) strategic interaction and (2) a more sensible objective function, let's now consider the empirical evidence provided in the paper about the optimality of strategies (Figure 3). First, there is no statistical test to demonstrate the simulation results (purple line in Fig. 3) and the actual countries' diversification levels are significantly different. Even though this should not be done since it is not methodologically correct, we can see in Figure 3 that simulation results are (almost) all in the gray area of the boxplots thus suggesting that the prediction of the model might be compatible with real world evidence, especially for research diversification. Second, I do not understand why the empirical fit is made on means and not on actual values. Fitting on binned data should be avoided when possible and a regression on all values should be provided. Also a statistical test for the inclusion of the squared term is needed, especially for research diversification there the relationship looks almost perfectly flat. Moreover, I would like to see the prediction of a random model in Figure 3, similarly to Figure 2(i). Finally, I don't

understand why a box is missing in Figure 3(b) and why the purple line goes down to 0 average degree of newly activated nodes for max diversification in subplot 3(a). We should know how exactly the purple lines have been obtained. Also no information is provided about the exact data sources which have been used for the networks of related products and research areas.

One more remark has to do with the sequence of future potential activations. In choosing with new opportunity to activate first, a perfectly informed country should consider also all the opportunities that would become available in the following steps. Conversely, it seems that based on the five diversification strategies considered in the paper countries just know about some local properties of their neighbors. I believe that the authors should make explicit their assumptions about the information available to decision makers in selecting the optimal diversification strategy in different networks.

Finally, there are some typos which need to be fixed (e.g. row 13 “therefore,requires”, row 17 “step We”; row 118 “). perform poorly”, row 119 “assymetric”, row 159 “Figure 2 h” should be “Figure 2 i”).

Reviewer #2 (Remarks to the Author):

The paper aims at going in depth in path dependent processes of knowledge diffusion by finding optimal strategies on new activities introduction, when activities are considered being related each other. Two key points are developed. First, the authors stress on the role of various structures of technological and scientific relatedness on the diffusion patterns, from wheel to scale-free networks. Second, they confront analytical and numerical results to real large dataset on knowledge and product spaces. Undoubtedly, the paper constitutes an original research that offers something new that was clearly missing in the growing research field of relatedness, diversification and economic growth, i.e. some theoretical and analytical foundations, as well as some strategic considerations on how and when to diversify at the country level.

The methodology does not suffer from criticisms and the balance between analytical and numerical simulations is perfectly justified. The data used for confronting results to real data on product and research spaces are well-known and constitute a reference in the community.

Then I consider the paper meets the quality standards of the journal. Only two remarks can be done, and could be taken into consideration for the final publication.

We thank Reviewer #2 for his time reviewing our paper. We are also glad for the positive assessment of our work and we hope he finds our comments and revised manuscript suitable for publication.

- First, the authors can suggest intuitions or announce further developments (and not demonstration) related to other network structures. The three structures considered in the paper are in a sense very limited. For instance, the structure of relatedness in science as well as in industry can present very different level of assortativity according to the different cultures in different countries. Considering assortative and non-assortative

structures can help better understanding knowledge diffusion and different patterns of diversification.

Thank you for raising this point. We agree that exploring the effects of other topological features, such as the assortativity of networks is an important question to consider. In our paper we start with simple network structures, to illuminate the basic intuition behind the strategies that maximize diffusion, and then move on to numerical simulations that can help us find near-optimal strategies for various heterogeneous topologies. Of course, the work is not exhaustive for all topologies, but after running many simulations we have reasons to believe that our numerical approach (combining greedy and majority strategies) are likely to provide near-optimal solutions for a wide range of heterogeneous networks (networks with hubs). So we leave the question of extending our results to all network topologies as one of the questions that can be motivated from this work.

Second, I think that the conclusion presents some confusing arguments. In particular, the reference to Marshall and agglomeration (and then, implicitly to Krugman on increasing returns and economic geography) seems to be unfitting with the topic under study. The product and knowledge spaces on which the authors work are completely disconnected from the spatial structure of countries. Exports and publications are simply considered as national aggregates, without any considerations on particular spatial structures such as cities or clusters. Then, the conclusion could be inappropriate for geographers. I would suggest making some changes in the conclusion, by referring more on researches related to economic policy in industry and science (what kind of public incentives to better target diversification strategy?)

Thank you for raising this point, we have revised the conclusion to clarify this point. We agree that the literature on agglomerations focuses on more narrow geographical units than countries,

such as cities and regions. Yet, many of the relatedness effects documented for cities and regions are also valid at the country scale (albeit a bit more weakly). The fact that the effects of relatedness are still observed at the country scale is probably due to the fact that countries are a meaningful unit of agglomeration, albeit not as meaningful as a city or region. We agree this point was confusing and have clarified the text in the manuscript to be more clear about why we consider countries as a coarse unit of economic agglomeration.

Pr. Jerome Vicente, University of Toulouse – France

We would, once again, like to express our thanks to Reviewer #2 for his time reviewing our manuscript, for his comments and positive assessment of our work.

Reviewer #3 (Remarks to the Author):

The authors address the interesting problem of knowledge diffusion as a complex contagion, and examine the effects of interdependency on optimal diffusion time. The overall idea is well-grounded and offers an important direction for research on productivity and knowledge networks.

We would like to thank Reviewer #3 for reviewing our paper. We are also glad for the positive assessment of our work. We hope the reviewer finds our comments and revised manuscript suitable for publication.

The authors could help to improve the clarity and reach of their project by making some changes in the exposition:

1) They would do well to foreground the important point that for knowledge diffusion, as

for all complex contagions, the role of hubs is determined by interdependencies with the peripheral nodes. This study demonstrates how these interdependencies shape the strategic role of hubs in accelerating knowledge diffusion (aka, “activation”).

We agree with the reviewer and have added text to emphasize this point, as suggested by the reviewer.

2) When α is first introduced, it would be useful if the authors provide empirical intuitions for situations where α is low, medium and high. This will add clarity to the theoretical and numerical analyses, which depend on this parameter.

We agree that this would help clarify the intuition and have added text to provide examples illustrating the interpretation of the values of α .

3) There is an abrupt transition in the exposition of the manuscript on the top of page 3, where the logic of the argument transitions from describing the interdependencies in activation dynamics in the empirical production and publication data, to framing the problem of when to activate a central node. There is a bit of a skip here –it does not follow naturally for the reader from the first two pages that “the strategic problem that we need to solve is to identify a sequence of targets that minimizes the total time needed to activate all nodes in a network.” Subsequently, it is clear that that the authors’ goal is to know when to activate areas of production in order to further increase (i.e., spread) production. However, the presentation here would benefit from a brief explanation of who faces this strategic problem, and what kinds of decisions actors are expected to make (e.g., activating hubs in the production network). This motivation would be helpful for connecting the theoretical and numerical analyses with the concluding data on empirical targeting decisions, and the lessons that these results have for actors who make these

decisions. As the authors struggle a bit to contend with this point in the final paragraph, it would be useful to anticipate this issue a bit earlier in the paper.

We agree with the reviewer and have added text in this transition to motivate the problem better.

4) It would be useful if the authors would provide some motivation for the five simple strategies they test. It is clear why they choose them: they use several familiar targeting strategies that are more or less representative of known techniques; however, it would be helpful if the authors could give a sense of the larger space that these targeting strategies explore. This would circumscribe the limits of their numerical explorations vis-à-vis their theoretical findings.

We agree with the reviewer and now motivate each of the five strategies used in the paper. We also agree that this circumscribes, and narrows, the scope of our empirical findings, which is good. The motivation for each of the five strategies is as follows. On the one hand, we have low-hanging fruit strategies, such as greedy and low-degree. These are strategies that target easy to activate nodes, but fail to consider the future strategic value of these nodes. On the other hand, we have more ambitious strategies, such as majority and high-degree, which target nodes that could be difficult to activate early on, but that can help activate other nodes later in the process. High degree strategies are particularly interesting as a benchmark because they are a common heuristic in problems of simple contagion. Finally, the random strategy allows us to describe what to expect in the absence of strategic choices, and hence, is an important null benchmark.

Additional minor points:

i) The text has numerous grammatical and syntactic errors, which can be easily fixed.

The intelligibility of the text would be greatly improved by editorial treatment.

Thank you, we have proofread the manuscript, and we hope the reviewer agree that the new resubmitted version is improved.

ii) This is not necessary, but the authors might consider using ‘activity diffusion’ instead of ‘knowledge diffusion’ since, as they discuss, there are many reinforcing factors (empirical path dependencies) that create complexity in the activation of nodes in a productivity network, and activation is not necessarily due to the spread of any particular kind of knowledge.

Thank you for this suggestion. We agree that our paper focuses on strategic activation. Yet, since our motivation is based on examples of the knowledge diffusion literature, and since this literature is vast, we prefer to stick to the conventional language.

Overall, the authors make a useful connection between several streams of research on complex contagions, knowledge diffusion, network structure, and productivity.

Thank you once more for the time reviewing our manuscript, for the useful comments that helped us improve the overall quality of our manuscript and for the positive assessment of our work. We hope the Reviewer find our manuscript suitable for publication.

Reviewer #4 (Remarks to the Author):

The paper analyses diffusion strategies in networks of related activities. A complex contagion model is developed to identify which activity should be targeted at each step. In the model, the probability that an agent will develop a new activity increases as a power function of active related activity with exponent α , as described in Equation 1. The

optimization problem is to minimize the total time needed to activate all nodes in the network (i.e. to develop all activities). An optimal solution to the problem is obtained for a wheel network (Eq. 3) whereas for other network topologies (generalized wheel and scale-free networks) numerical results are provided. The optimal strategy mixes the greedy and the majority strategies. The optimal mix favors majority for low values of α and the greedy strategy when α is high enough. Results are validated by looking at the strategies followed by countries in the development of new exports and research areas.

We would like to thank the Reviewer #4 for the time reviewing our manuscript and for his assessment. Below we provide a point by point answer to the questions and remarks raised.

I have a few major concerns about the paper and there are some other minor issues that should be addressed by the authors, especially in the empirical part of the manuscript.

My comments are as follows:

1) Strategic interaction is not considered. Agents develop a new activity if they outperform rivals ($RCA \text{ per capita} > 1/4$ and $RCA \text{ per capita} > 1$ for research and exports, respectively). RCA depends on the strategies of other agents. As it is evident in Eq. 14 in the appendix, RCA of country c is a function of exports of other countries c' . As described in Fig. 1(f) a potentially active product is connected to active nodes in the product space. By construction, connections are present when a sufficiently large number of countries jointly export active and potentially active products, with $RCA > 1$. This implies that the optimal strategy of country c depends on the strategy of other agents who are likely to develop the same product. For instance, if all of them target the same potentially active node by increasing per capita exports to the same extent, their efforts are likely to cancel out. Neglecting to consider strategic interaction in this setting is a major limitation of the paper. More in general, strategic interaction is core in game-theoretic competition models widely applied to international studies.

We agree that strategic interactions are a fascinating topic of study. In fact, they have been at the core of many discussions among the team of authors. The reason why we decided not to include strategic interactions in the current manuscript, is that these would require describing a large additional set of possible assumptions (introducing a second model), which would elongate the paper and scatter its focus. For instance, if two countries were to target and enter

5
the same product, what should their market participation be? Should we distribute that evenly among them? Or would the countries that have more relatedness enjoy a larger market share? If that is something that the economic actors knew, would they then change their choices to incorporate not only the probability of succeeding at an activity, but also, their expected market share? How would we model the effect of these new entries on the incumbents? Would incumbents leave a product at some point? If so, when?

We wholeheartedly agree with the reviewer that the question of strategic interaction in the context of strategic diffusion is fascinating. Yet, given the large number of possible assumptions that one could bring to such strategic interactions, we believe that the community is best served if the problem of strategic interaction in strategic diffusion is considered in a future manuscript, by us or by other authors, motivated by this one.

Also, the way in which the RCA thresholds are fixed is totally arbitrary. A justification is needed of why we should consider as a relevant transition when RCA goes from lower than 0.25 to an RCA greater than 0.25 in the product space.

We agree that the RCA threshold values used are somehow arbitrary. Yet, our results are robust to these thresholds. We use a threshold RCA of 0.25 threshold since this has been used previously in the literature (1) and (2). At this point, the population based threshold of RCA provides a network of countries and products with a similar density than the one of exports based RCA when we set the RCA threshold equal to 1.

[1] Felipe, J. & Hidalgo, C. Economic diversification implications for kazakhstan. *Development and Modern Industrial Policy in Practice: Issues and Country Experiences*; Felipe, J., Ed 160-196 (2015).

[2] Bustos, Sebastián, et al. "The dynamics of nestedness predicts the evolution of industrial ecosystems." *PloS one* 7.11 (2012): e49393.

We have clarified this choice in the manuscript.

2) Diversification is not the objective. In the paper, the authors implicitly assume that all activities (products, publications) are of the same value. Evidently, this is not the case. Thus, the objective function is questionable. It is much more sensible to assume that countries' objective in trade is to maximize the contribution to GDP (or the trade balance) per capita (total exports per capita – total imports per capita). Total exports is the sum across all exported products (extensive margin) of the dollar value of exported goods (intensive margin). Thus, countries can develop strategies to export less goods but at a higher value. Indeed, it is well know that most developed countries export less products than mid-income countries, and no country exports all goods. The same trade-off applies to scientific production (less papers of higher impact). In sum, we cannot take for granted that maximal diversification is good. Indeed, policies in Europe and elsewhere are meant to pursue smart specialization, not necessarily as a way to minimize the time to maximal diversification.

We agree with the reviewer that diversification is not the only objective. In fact, one could define multiple objective functions (and many people do). Some of these functions are maximizing GDP, or GDP per capita (which is what the reviewer suggests). Others involve minimizing income inequality, maximizing economic complexity, or reducing the carbon footprint of an economy. In principle, our approach could be used to model these different objective functions. Of course, the strategies needed to maximize these alternative functions would not be identical

to the ones needed to maximize diversity, but a similar approach to the one presented in our paper could be used for that purpose. For instance, we could associate each product to GDP by combining the total market size of a product with the size of that country's export economy. We could also associate each product to a carbon footprint, by using data of the energy and carbon intensity of the industries associated with those products. We agree that our exposition was unclear, and could be interpreted as proposing diversification as the only objective. We have now clarified the wording in our paper, to communicate that the method can be used for different objective functions, but that we focus on diversification as a baseline because most objective functions will require traversing the networks of related activities, which is what the diversification function tries to do as fast as possible. Moreover, other objective functions, like the ones described above, would involve assigning a value to each node. In the case of diversification, this value is equal to one for all nodes, making this the quintessential neutral baseline, but of course, it is not the only objective function that one could explore using this method.

3) Empirical results are weak. The authors claim that countries follow strategies that are slightly sub-optimal. Having said that we cannot discard the possibility that countries are just more sophisticated and they follow an optimal strategy which considers (1) strategic interaction and (2) a more sensible objective function,

We agree, and hence qualify the suboptimality as a narrow result that is in respect to the diversification objective function, not all possible objective functions. We apologize if this was unclear and have clarified it in the present manuscript.

let's now consider the empirical evidence provided in the paper about the optimality of strategies (Figure 3). First, there is no statistical test to demonstrate the simulation results (purple line in Fig. 3) and the actual countries' diversification levels are significantly different.

Even though this should not be done since it is not methodologically correct, we can see in Figure 3 that simulation results are (almost) all in the gray area of the boxplots thus suggesting that the prediction of the model might be compatible with real world evidence, especially for research diversification. Second, I do not understand why the empirical fit is made on means and not on actual values. Fitting on binned data should be avoided when possible and a regression on all values should be provided.

We apologize if this was confusing. The fits were not performed over the fitted values, but over the entire dataset. It was our mistake to have said that in the caption of Figure 3. We have now corrected the manuscript.

Also a statistical test for the inclusion of the squared term is needed, especially for research diversification there the relationship looks almost perfectly flat. Moreover, I would like to see the prediction of a random model in Figure 3, similarly to Figure 2(i).

We have added a null model baseline and statistical test for the significance of the square term. The significance levels are indicated with * below each coefficient. All coefficients, including those of square terms, are significant.

Finally, I don't understand why a box is missing in Figure 3(b)

The reason why one box is missing is that there were no events in that bin. The empirical data is sparse, so we removed the bins with no events in the observation window.

and why the purple line goes down to 0 average degree of newly activated nodes for max diversification in subplot 3(a).

We agree this is not obvious. The reason why the average degree approaches low values ([1-10]) for max diversification in the optimal strategy is a direct consequence of the theory. To maximize the speed of diversification one needs to target high and medium connectivity nodes in the beginning and leave leaf nodes (nodes of degree one) for the end. That's because nodes of degree one cannot be activated if their only neighbor is not active, and hence, cannot contribute to the activation of any additional nodes. So the optimal strategy to minimize total diversification time involves activating nodes of degree one at the end of the process (when diversification is near its maximum).

We should know how exactly the purple lines have been obtained.

Purple lines were obtained by applying the optimal strategy identified through numerical simulations combining the greedy and majority strategies. We now make this more clear in the main text.

Also no information is provided about the exact data sources which have been used for the networks of related products and research areas.

Information about the exact data sources used for the networks is presented in the supplementary material section 5 (for the product space) and section 6 (for the research space) and the original sources (both networks have been previously published). Since both networks have been previously published, we think that describing them again in the main text of our paper is unnecessary, and that's why we leave their description for supplementary information.

One more remark has to do with the sequence of future potential activations. In choosing with new opportunity to activate first, a perfectly informed country should consider also all the opportunities that would become available in the following steps. Conversely, it seems that based on the five diversification strategies considered in the paper countries

just know about some local properties of their neighbors. I believe that the authors should make explicit their assumptions about the information available to decision makers in selecting the optimal diversification strategy in different networks.

Thank you for point the lack of clearness in our work about this point, we have clarified the assumptions of each strategy on the information available to each agent. We consider that agents (countries) are rationally bounded and unable to assess the full topology of the network, which we believe better translates to many real life scenarios.

Finally, there are some typos which need to be fixed (e.g. row 13 “therefore,requires”, row 17 “step We”; row 118 “). perform poorly”, row 119 “assymetric”, row 159 “Figure 2 h” should be “Figure 2 i”).

Thanks for pointing to these typos. We have corrected in the new version.

Reviewers' comments:

Reviewer #2 (Remarks to the Author):

The changes done in the second version (echoing other network structural properties, and clarifying national/regional agglomerative pressures of knowledge diffusion) meet my expectations. I now recommend the paper for publication. Jerome Vicente (Sciences-Po Toulouse)

Reviewer #3 (Remarks to the Author):

The authors should include a reference to Centola (2010) when discussing complex contagions. Other than that the paper is suitable for publication.

Reviewer #4 (Remarks to the Author):

The authors have improved the paper in the new version. However, I'm still not satisfied of the way the authors addressed my main concern, that is to say the role of competition (or strategic interdependence) in the model.

In the reply letter they essentially say they won't consider this issue in the present paper but in future work. This is not fine to me since in the empirical part of the paper they use a measure of Revealed Competitive Advantage (RCA). How to predict competitive advantage by a model in which competition is explicitly ruled out?

There is still some confusion about say: a) start exporting a good and b) passing a certain threshold of RCA. By definition, RCA depends on the export performance of other countries. Thus, inevitably, if the activation of a new node implies to increase RCA above a certain threshold, since that threshold depends on exports by others, players must take onboard that others will do. Consider the simple

case of only two players. If they both target the same node by increasing export of the same amount, their efforts will cancel out and RCA won't change. In such a setting, not to consider that rivals will do is simply not possible, since trade is intrinsically a competitive phenomenon and RCA has a measure is meant to capture that (the acronym stays for Revealed Competitive Advantage!). To avoid the issue of strategic interdependence, which is severe, the authors must at least reframe the model as monopolistic competition and change the empirical measures accordingly. Also, to better justify the objective function in the paper, I think the authors should mention at least one example of a country which has announced as a goal to increase the number of exported goods.

Reviewer#4

The authors have improved the paper in the new version. However, I'm still not satisfied of the way the authors addressed my main concern, that is to say the role of competition (or strategic interdependence) in the model.

In the reply letter they essentially say they won't consider this issue in the present paper but in future work. This is not fine to me since in the empirical part of the paper they use a measure of Revealed Competitive Advantage (RCA). How to predict competitive advantage by a model in which competition is explicitly ruled out? There is still some confusion about say: a) start exporting a good and b) passing a certain threshold of RCA. By definition, RCA depends on the export performance of other countries. Thus, inevitably, if the activation of a new node implies to increase RCA above a certain threshold, since that threshold depends on exports by others, players must take onboard that others will do. Consider the simple case of only two players. If they both target the same node by increasing export of the same amount, their efforts will cancel out and RCA won't change. In such a setting, not to consider that rivals will do is simply not possible, since trade is intrinsically a competitive phenomenon and RCA has a measure is meant to capture that (the acronym stays for Revealed Competitive Advantage!). To avoid the issue of strategic interdependence, which is severe, the authors must at least reframe the model as monopolistic competition and change the empirical measures accordingly. Also, to better justify the objective function in the paper, I think the authors should mention at least one example of a country which has announced as a goal to increase the number of exported goods.

We appreciate the reviewer for taking time to look at our paper once again. We believe, however, that there may be a misunderstanding about the assumptions in our work, and about the mathematics of RCA (Revealed Comparative Advantage). Our hope is to help solve this misunderstanding.

First, our paper uses what is a standard assumption in economic development. This is that world markets are relatively large compared to domestic markets. So the main problem that countries face, when entering a new product or research activity, is not that of competing with other new entrants, but that of capturing a small market share in a world of large incumbents. The idea that the world markets of products are big enough to absorb small shocks on the supply side is not only a standard assumption, but also, consistent with the data. On average when a country enters a new product in the product space (reaches **Revealed Comparative Advantage** $RCA > 1$) it is exporting less than 1% of global trade in that product (we calculated this using historical trade data). This tells us that entry events are miniscule in trade volume (compared to the market size of goods), and hence, it is reasonable to assume that the global market will be able to accommodate a new entrant. More to the point it shows that what new entrants should worry about, is not other new entrants, but existing incumbents. In most products, incumbents account for much more than 90% of all exports (more like 99%) in a given year, and they are the true competition that entrants need to overcome to develop the minuscule market share they need to develop RCA in a new product. So the true strategic game is the opposite of what the reviewer suggests. It is not so much a game where new entrants compete against each other, but a game where new entrants compete against gigantic incumbents.

Second, this rationale also extends to the definition of RCA. Since new entrants account usually for less than 1% of the total exports in a product (even when they are successful), they do not really move the needle in the denominator of the definition of RCA, and the effects of two entrants, do not cancel out (as the reviewer suggests).

Consider two new entrants (c and c'), with exports in product p that increase by E_c and $E_{c'}$. Hence, the total exports of these countries in product p change from $X_{cp} \rightarrow X_{cp} + E_c$ and from $X_{c'p} \rightarrow X_{c'p} + E_{c'}$.

Now, let's remember the definition of RCA we are using, which is population based:

$$RCA_{cp} = \frac{X_{cp}}{N_c} / \frac{\sum_{c'} X_{c'p}}{\sum_{c'} N_{c'}}$$

Here N_c is the population of country c . Since increases in exports do not change population, the terms that depend on N are constant and we can ignore them to simplify the algebra. Hence, we

only need to worry about changes in X_{cp}/X_p , where X_p is the total market of a product (X_{cp} summed over c).

Now, if only the first country (c) enters this ratio grows to:

$$(X_{cp}+E_c)/(X_p+E_c)$$

or

$$X_{cp}(1+E_c/X_{cp})/(X_p(1+E_c/X_p))$$

We now know this is larger than before because the increase of exports of a country in a product represent a larger share of the previous export of that country in that product (E_c/X_{cp}) than of the world market in that product (E_c/X_p) ($E_c/X_{cp} \gg E_c/X_p$ since $X_{cp} \ll X_p$).

Now, the reviewer suggest that if the second country enters this effect will cancel out (and hence this ratio should go back to X_{cp}/X_p). But this is not true. When the second country enters we obtain:

$$(X_{cp}+E_c)/(X_p+E_c+E_{c'})$$

Which can be simplified to:

$$(X_{cp}+E_c)/((X_p+E_c)(1+(E_{c'}/(X_p+E_c))))$$

The correction imposed by the entry of the second entrant is of the order of $E_{c'}/(X_p+E_c)$ in the denominator, which we know is less than 1% (since $E_{c'}/X_p$ is already less than 1% on average, and certainly, much smaller than E_c/X_{cp}). So the effects do not cancel out as the reviewer hypothesized (the effect of the second entrant is negligible because the incumbents explain most of the denominator).

Third, the reviewer ignores the application of the paper to the research space. The paper looks in parallel at countries entering export products and research activities. In the case of research activities, the assumption that the world market for research is relatively large is also valid. But there is more. In the case of research activities one could easily argue that competition creates an incentive to enter a new research activity since researchers want to enter areas that are growing and have a strong international community. These areas will have more citations, funding, and colleagues, so the strategic choice here is to enter the areas that others are entering (not to avoid them). In any case, we believe these strategic interactions are second order, and also, that the knowledge that scholars have about what other scholars are working on at a planetary scale is at best, highly imperfect, and most certainly incomplete.

Also, the reviewer asks us to name a country that has mentioned increasing the diversity of their economies as one of their development goals. In fact, increasing the diversity and sophistication

of exports is a common economic goal development that many countries have made explicit in their strategic policy documents. Here are five countries that have made economic diversity an explicit development target, but of course, there are more.

Saudi Arabia has explicitly included increasing economic diversity as one of their primary development goals, as stated in their Official Economic Development Strategy (Vision 2030). Here is a guardian article reporting on the Saudi plan:

<https://www.theguardian.com/world/2016/apr/25/saudi-arabia-approves-ambitious-plan-to-move-economy-beyond-oil>

and here is the official document published by The Kingdom of Saudi Arabia.

<http://vision2030.gov.sa/download/file/fid/417>

The Saudi effort is so determined, that it even plans to transform Aramco (the largest oil company in the world) into a more diversified industrial conglomerate.

Peru is another country that has an explicit Plan for Economic Diversification, since they identified this as an strategic goal.

<http://sinia.minam.gob.pe/documentos/plan-nacional-diversificacion-productiva>

Chile is another example. It's innovation strategy is built on top of four pillars, the second one being to diversify its productive structure.

<http://www.economia.gob.cl/wp-content/uploads/2014/12/Plan-Nacional-de-Innovaci%C3%B3n1.pdf>

Qatar also has the goal of diversifying its economy,

<https://oxfordbusinessgroup.com/overview/broader-base-government-making-efforts-diversify-economy>, and the same is true for Indonesia:

<http://www.channelnewsasia.com/news/business/indonesia-seeks-to-diversify-economy-as-commodity-exports-take-a-8229884>

<https://www.pressreader.com/indonesia/the-jakarta-post/20170714/281973197697252>

We hope this clarifications help resolve any misunderstanding. We have added text in our manuscript to make our assumptions more explicit, and also, to mention explicitly some of the countries that have the diversification of their economies as one of their primary goals.

REVIEWERS' COMMENTS:

Reviewer #4 (Remarks to the Author):

I think the authors did a good job in revising the paper and I do not have other comments